# Recent Advances in Anode Materials for Sodium-Ion Batteries

**Xue Bai** [1,2,*], **Nannan Wu** [1], **Gengchen Yu** [1] **and Tao Li** [3,*]

1 School of Materials Science and Engineering, Shandong University of Science and Technology, Qingdao 266590, China; skd996364@sdust.edu.cn (N.W.); ygc260366@163.com (G.Y.)
2 Key Laboratory of Advanced Energy Materials Chemistry (Ministry of Education), Nankai University, Tianjin 300071, China
3 Key Laboratory of Liquid-Solid Structural Evolution and Processing of Materials of Ministry of Education, Shandong University, Jinan 250061, China
* Correspondence: xbai@sdust.edu.cn (X.B.); tao.li@sdu.edu.cn (T.L.)

**Abstract:** Although lithium-ion battery (LIB) technology has prevailed for years, the growing pressure and increased cost of lithium sources urge the rapid development of other promising energy storage devices. As a low-cost alternative, sodium-ion batteries (SIBs) with similar properties of electrochemical reaction have caught researchers' attention. Nevertheless, great challenges of inferior reversible capacity and poor lifespan induced by the bigger ionic radius of sodium ions still exist. To solve these problems, improvements to anode materials prove to be an effective way. Herein, the latest research on promising anodes in SIBs is summarized, and the further prospects are also illustrated.

**Keywords:** sodium-ion batteries; anode; carbon; metal oxides; alloy

## 1. Introduction

Given the growing concerns about environmental pollution and energy shortages, renewable and clean energy, for instance, solar, wind, and tidal energy, has been regarded as the most prominent alternative candidate to solve future survival issues. Nowadays, the intermittent period problem still hinders the full utilization of these new energy sources, which cause extreme pressure on electrical energy storage systems (EES) [1–3]. Moreover, the current transportation systems, which are usually powered by fossil issues, would be more promising if they were replaced by electrically driven devices. Batteries, especially lithium-ion batteries (LIBs) with intriguing eco-friendly features and a high conversion efficiency, have occupied the industry market over the past decades [4]. However, due to the increased demand for LIBs, the availability of lithium sources has decreased, and the cost of lithium has inevitably risen [5]. Therefore, it is necessary to explore new energy storage techniques to meet this urgent need in the near future.

Inspired by the similar electrochemical reaction mechanisms to LIBs, sodium-ion batteries (SIBs), which are another promising power technology, could also be applied as energy storage devices for both large-scale energy storage and smart grid applications [6]. In addition, SIBs also exhibit outstanding merits in contrast to lithium, thanks to their high abundance and low cost of sodium sources. As reported for a 7 kW/11.5 kWh LMO-synthetic graphite battery, 3.8% of the cost for electrode materials (USD 1022) as well as 1.3% of the total cost for complete battery (USD 2981) could be achieved if the same amount of lithium was replaced by sodium [7]. However, a larger ionic radius of 1.02 Å for Na$^+$ (0.76 Å for Li$^+$) usually leads to slower reaction kinetics, which causes an inferior reversible capacity and a poor rate capability [8]. Anode materials, which are one of the key components in LIBs, still exhibit their importance in promoting the development of SIBs. There are three mechanisms that can store lithium ions, including intercalation/de-intercalation, the conversion method, and the alloying/de-alloying reaction [9]. Although many kinds of materials are suitable for serving the positive electrodes of SIBs, by contrast, there seem to be few available anode candidates. Similarly to Li anodes in LIBs, Na foil

is not an advisable option for SIBs due to their high reactivity, dendrite formation, and instability in most organic electrolytes. And its melting point is as low as 97.7 °C, causing a safety risk for SIBs as well. So far, several main types of anode materials have been under investigation, including (1) carbonaceous materials, (2) metal oxides, (3) alloys, (4) sulfides and phosphates, (5) organic compounds, and (6) 2D materials. Thus, this review will summarize the recent research progress on SIB anodes and the corresponding electrochemical behaviors in detail.

## 2. Anode Materials

### 2.1. Carbon

#### 2.1.1. Graphite

Carbonaceous-based anode materials have been commonly applied in LIBs for decades [10]. Graphite, when applied in LIBs, is capable of releasing a high theoretical reversible capability (372 mA h g$^{-1}$). However, unlike the application in LIBs, Na$^+$ can hardly electrochemically intercalate into graphite layers because of its larger radius, so it is accepted that graphite does not exhibit a desirable intercalation ability for Na$^+$. Theoretical calculations reveal that a critical minimum distance of 0.37 nm between the graphitic layers is essential to achieve facile Na$^+$ transfer [11], but graphite provides an insufficient interlayer spacing of 0.34 nm. Thus, the common graphite is not as competent as a SIB anode without further modification.

One effective method to improve the sodium storage performance of graphite is to increase the interlayer spacing. Figure 1a schematically depicts the Na$^+$ storage in graphite-based anodes [12]. Wen et al. adopted a two-step oxidation–reduction process to easily fabricate expanded graphite. The interlayer space could be manipulated easily via this simple method, and the modified graphite exhibited an interlayer distance of 0.43 nm. The structural robustness observed via in situ TEM indicates that the interlayer change was reproducible during the multiple charge/discharge processes (Figure 1b). Thus, the expanded graphite could reach a high specific capacity of 284 mA h g$^{-1}$ at 20 mA g$^{-1}$ and 184 mA h g$^{-1}$ at 100 mA g$^{-1}$ (Figure 1c), and undergo 2000 cycles at 20 mA g$^{-1}$ (Figure 1d).

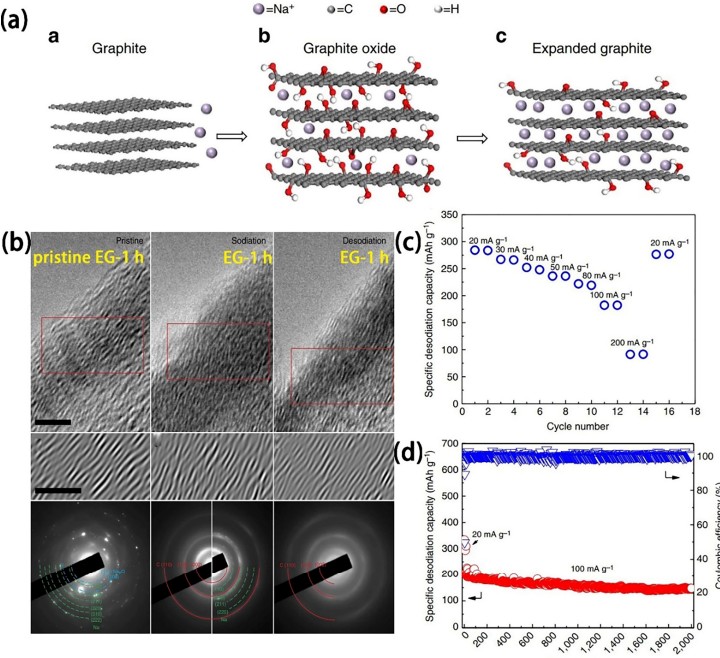

**Figure 1.** (**a**) Schematic illustration of sodium storage and (**b**) in situ TEM investigation of sodium storage mechanism in graphite-based materials. (**c**) Rate capability test and (**d**) long-term cycling stability at 100 mA g$^{-1}$ of expanded graphite. Reproduced with permission [12]. Copyright 2014, Springer Nature.

The other common way to push $Na^+$ into a graphite interlayer depends on the co-intercalation phenomenon of $Na^+$ in some specific solvents. In terms of LIBs, $Li_xC_6$ ($0 < x \leqslant 1$), also called graphite intercalation compounds (GICs), is the main product after $Li^+$ intercalation. Even though $K^+$ has a larger radius than $Na^+$, it is capable of inserting into graphite and reversibly extracting from graphite by forming K-GICs ($C_8$) in K-ion batteries (KIBs) [13]. However, the weak $Na^+-$graphene cation$-\pi$ interaction leads to difficult $Na^+$ storage in graphite, namely $Na_xC$ ($x \approx 0$), and no stable Na-C binary compounds have been identified so far. Nevertheless, with the assistance of ether-based solvents, graphite could reversibly accommodate $Na^+$ [14], and it is a more effective approach to adopt graphite as an SIB anode than functionalizing graphite. The intercalation of solvated $Na^+$ could be illustrated as the following equation (also applied to other alkali metals) [15]: $C_n + e^- + Na^+ + y\,(solv.) \rightleftarrows Na^+(solv.)_yC^-_n$, in which $C_n$ is n carbon atoms in the graphite lattice and solv. represents the solvent molecule. The ether-based solvents, especially linear ether, containing multiple O atoms are able to capture one $Na^+$ ion promptly, leading to further and stronger solvation of $Na^+$ ions [16].

Kim and coworkers carried out a series of investigations on the co-intercalation phenomenon of alkali ions to compare the graphite electrode utilized in KIBs, LIBs, and SIBs [17]. Half-cells with 1 M $MCF_3SO_3$ (M = Li, Na, or K) in diethylene glycol dimethyl ether (DEGDME) electrolyte were used to examine the electrochemical properties. Of particular interest, the potentials of over 1 V for $K^+$ and $Li^+$ are quite different from that in conventional KIBs (0.2 V vs. $K/K^+$) or LIBs (0.1 V vs. $Li/Li^+$), indicating significant dissimilar reaction mechanisms. From the plots of voltage vs. NHE, regardless of the ionization energy, the average voltage is proportional to the ion size, implying that Na and K are more likely to insert into graphite than Li, which is in contrast with what is usually believed. Moreover, the calculated interlayer distance of each intermediate phase reveals an increasing value in the order of Li, Na, and K. The larger solvated ion graphite compound results in reduced repulsion between negatively charged graphene layers in the discharged state as well as a higher alkali ion storage potential.

The study using $^2H$ solid-state NMR indicated that in Na-diglyme-GICs complexes, two diglyme molecules coordinate to each $Na^+$ ion weakly via the central oxygen atom in diglyme and rotate around an axis along O-Na at an ambient temperature [18]. Furthermore, first-principle molecular dynamics simulations also evidenced that $Na^+$ ions are able to insert into graphite easily with the aid of diglyme electrolyte [19]. For $Na^+-$diglyme co-intercalation, the diglyme-graphene van der Waals interaction played an important role in stabilizing the mechanical integrity by enhancing the coupling strength between graphene interlayers, and the fully solvated $Na^+$ ions could diffuse in/out the graphite interlayers with ease. Thus, the fast $Na^+$ movement contributed to an even higher $Na^+-$diglyme conductivity, which was almost five orders of magnitude higher than that of the Li counterpart.

Understanding the co-intercalation behavior of $Na^+$ in the presence of ether-based electrolytes and fabricating SIB full cells with desirable electrochemical reactions is feasible. A full cell consisting of $Na_{1.5}VPO_{4.8}F_{0.7}$ positive electrodes and natural graphite anodes with a weight ratio of 1.5: 1.0 was under evaluation [20]. The graphite//1 M $NaPF_6$ in DEGDME//$Na_{1.5}VPO_{4.8}F_{0.7}$ cell delivered a specific capacity of over 100 mA h $g^{-1}_{anode}$ and 41 mA h $g^{-1}_{electrode}$ at 0.1 A $g^{-1}$, and the average discharge voltage occurred at around 2.92 V, corresponding to an energy density of $\approx$120 W h $kg^{-1}$. After cycling 250 cycles at 0.5 A $g^{-1}$, 70% of the initial discharge capacity was still maintained. Hasa addressed an optimized SIB full battery with a graphite anode [21]. The graphite anode was activated in a half-cell (1 M TEGDME-$NaClO_4$) to eliminate the initial irreversible specific capacity. The electrochemical performance of graphite//1 M $NaClO_4$ in TEGDME //$Na_{0.7}CoO_2$ full SIB was tested between 3.7 and 0.5 V. A specific capacity of 80 mA h $g^{-1}$ was delivered upon 100 cycles at 175 mA $g^{-1}$ and the stable long cycling life reached more than 1200 cycles, even at 1750 mA $g^{-1}$.

### 2.1.2. Non-Graphite Carbon

Hard carbon, an important kind of non-graphite carbon, usually synthesizes at high temperatures of over 1000 °C [22]. Stevens and coworkers have reported pioneering work on the electrochemical behavior of hard carbon in SIBs [23]. They used glucose as a carbon precursor, carbonized it at 1000 °C for the assessment, and then obtained hard carbon that delivered a reversible specific capacity of ca. 300 mA h g$^{-1}$. As for the potential profiles, the sloping region below 1 V is related to Na ion intercalation between graphene sheets, and the following voltage intercalation reaction platform below 100 mV is induced by Na ions filling in the pores. It is found that a similar reaction mechanism to that of LIBs occurs during Na$^+$ insertion. The random stacking of carbon fragments with a lateral extent of about 40 Å causes small regions where some of the layers are parallel to each other and act as a house of cards. Recently, Z. Lu's group reported the extra sodiation behavior of hard carbon [24]. They claimed that excess sodium storage sites with stable C-N• and C-C• radicals would be provided by introducing pyridine N during cycling. Various in situ and ex situ characterizations further verified that extra sodiation sites come from the electrostatic interaction at low potentials. Finally, the hard carbon with pyridine N doping reveals outstanding high specific capacity of 434 mA h g$^{-1}$ at 20 mA g$^{-1}$ together with a high capacity retention of 98.7% after 5000 cycles. Thus, it appears that hard carbon is very suitable for rechargeable SIBs.

Notable, all the modification methods adopted in LIBs are applied to SIBs, mostly including heteroatom doping and morphology design [25]. Doping with light-weight atoms (such as B, N, P, and S) is conductive to modifying the local bonding environment, expanding the interlayer distance, and improving the electron distribution on the carbon surface, thus contributing to the enhancement of electronic and ionic transfer [26]. Recently, C. Liu's team prepared a sulfur-doped camphor tree-derived HC anode [27]. Successful S doping could lead to increased interlayer spacing and facilitate the intercalation/deintercalation of Na$^+$ ions. Thus, the Na half-cell exhibited a discharge-specific capacity of 616.7 mA h g$^{-1}$ with an ICE of 66.61%. Additionally, a reversible specific capacity of 145.6 mA h g$^{-1}$ over 500 cycles at 2000 mA g$^{-1}$ was achieved.

In addition, high-performance batteries obtained by modifying structures are easily attainable [28]. For example, 1D carbon nanofibers fabricated by electrospinning followed by thermal annealing demonstrated a specific capacity of 248 mA h g$^{-1}$ after 100 cycles at 100 mA g$^{-1}$ with 91% capacity retention [29]. Bio-based carbon fibers derived from polyacrylonitrile (PAN)-humic acid (HA) exhibited a specific capacity of 249.6 mA h g$^{-1}$ at 0.1 A g$^{-1}$ over 100 cycles and 81.7 mA h g$^{-1}$ at 1 A g$^{-1}$ [30]. The S and N co-doped hollow carbon spheres denoted a specific capacity of 110 mA h g$^{-1}$ at 10 A g$^{-1}$, and only 0.0195 mA h g$^{-1}$ per cycle faded over 2000 cycles [31].

When served as a negative electrode in a full battery, hard carbon is the best choice, owing to its high capacity, low working potential, and stable long-term cycling properties. The full batteries consisting of NaCrO$_2$ cathodes and hard carbon anodes exhibited an excellent performance even at 10 A g$^{-1}$ [32], and the Na$_{3+x}$V$_2$(PO$_4$)$_2$F$_3$/hard carbon system could deliver an energy density of 265 W h kg$^{-1}$ [33]. Since hard carbon suffers from a high initial irreversible capacity, pre-sodiation is necessary. Pretreatment of an anode helps to extract the maximal capacity and retain structural stability. Llave et al. studied the effect of pretreated anodes in full SIBs and LIBs. The superior capacities of pretreated hard carbon compared to the untreated ones, either in SIBs or in LIBs, demonstrate the significance of the pretreatment for obtaining maximal capacity in the full cells [34]. Similarly, when adopting P2/P3/O2-Na$_{0.76}$Mn$_{0.5}$Ni$_{0.3}$Fe$_{0.1}$Mg$_{0.1}$O$_2$ as the cathode, the well-balanced cells with hard carbon anodes, which have been subjected to pre-activation, displayed an energy density of about 240 W h kg$^{-1}_{cathode+anode}$ at 0.1 C, while the non-presodiated anodes delivered a lower specific discharge energy of 210 W h kg$^{-1}$ at the same current density [35].

For years, the most common way to obtain hard carbons has been by pyrolyzing different precursors such as organic polymers, hydrocarbons, and saccharides [36,37]. In the last few years, biomass wastes, such as diverse plant peels [38–41], prawn shells [42],

natural cotton [43], coconut oil [44], and oatmeal [45], have been explored as anodes in SIBs as a kind of renewable source. Since over a million tons of these materials are discarded without any utilization, taking advantage of them facilitates reductions in waste and lowers the cost of energy storage packs. Wu et al. prepared hard carbon derived from wild apples [46], which underwent a simple drying process in an oven and a subsequent strict dehydration process in the presence of phosphoric acid. The aim of adding phosphoric acid as an activation agent was to form porous carbon with a high specific surface area. The hard carbon//1M $NaClO_4$ in EC/PC//P2-$Na_xNi_{0.22}Co_{0.11}Mn_{0.66}O_2$ cell revealed a stable cycling performance with a high specific capacity of 250 mA h g$^{-1}$ after cycling 100 times between 3.95 and 0.5 V at 0.1 C, as well as 220 mA h g$^{-1}$ at 1 C and 183 mA h g$^{-1}$ at 2 C. They claimed that the observed increasing cell polarization was mainly due to the un-optimized P2-$Na_xNi_{0.22}Co_{0.11}Mn_{0.66}O_2$ positive electrode, and that the biomass carbon was a potential material suitable for anode materials. Apart from hard carbons, some soft carbons, including anthracite and pitch, have been reported as well. [47,48]

In brief, carbonaceous materials with nontoxicity, abundance, and a satisfactory capacity in the range of 200–500 mA h g$^{-1}$ are good options for anodes of SIBs. By means of increasing the interlayer distance or the co-intercalation effect, graphite could successfully achieve the insertion/exaction of $Na^+$ and deliver a desirable discharge capacity. Usually, amorphous carbon with large porosity provides higher capacity than graphite because of the favorable sodium storage in surface defects. However, it should be noted that the initial Coulombic efficiency is too low for carbonaceous materials, and activation in a half-cell to remove the initial irreversible capacity prior to applying in full cells is favorable to compensate for partial sodium loss. How to improve the initial Coulombic efficiency (ICE) is the key job in the next study. Converting biomass wastes to carbon as raw materials for batteries not only achieves resource recycling but also benefits the environment. Thus, there is great potential for carbon's commercialization with further enhanced electrochemical performance.

### 2.2. Metal Oxides

2.2.1. Titanium-Based Oxides

Among the diverse kinds of metal oxides for SIB anodes, titanium-based oxides based on intercalation reactions are the most studied candidates, mainly including $Li_4Ti_5O_{12}$ (LTO), $TiO_2$, and Na-Ti-O composites.

LTO has been regarded as an anode material for LIBs for many years due to its merits of non-toxicity, safety, and low cost. Especially, the negligible structural change is especially attractive, not only for LIBs but also for SIBs [49,50]. Compared with LTO, $TiO_2$ has the distinct advantage of being much lower in cost due to the absence of Li. According to the different arrangements of $TiO_6$ octahedra, $TiO_2$ could be classified into four polymorphisms: anatase, rutile, bronze ($TiO_2$-B), and brookite. The anatase phase has been found to be more active than rutile because of the inherent 2D channels, and $TiO_2$-B reveals the highest practical capacity in the family of $TiO_2$ because of its strong storage capability to store 1 $Li^+$ per Ti. Even though $TiO_2$ exhibits stable long-cycling properties, the low theoretical discharge-specific capacity of 335 mA h g$^{-1}$ and inherent poor electrical conductivity are still big challenges. Meanwhile, the high potential of 1.75 V vs. Li/$Li^+$ requires a cathode with a high voltage once used in full batteries. Xiong et al. fabricated an amorphous $TiO_2$ nanotube ($TiO_2$NT) to behave as an SIB anode in 2011 [51]. The $TiO_2$NT was directly grown on the Ti current collector, avoiding the complexity of adding binders and additives during cell assembly. Coupled with the $Na_{1.0}Li_{0.2}Ni_{0.25}Mn_{0.75}O_\delta$ cathode, the obtained full cell, which was tested in a voltage window of 1.0–2.6 V, exhibited a discharge-specific capacity of about 80 mA h g$^{-1}$ at 11 mA g$^{-1}$ with an operation voltage of 1.8 V.

In order to improve the intrinsic low conductivity, nanoscale $TiO_2$ with diverse morphologies and modifications by coating and doping are required [52,53]. Yolk–shell $TiO_2$@C presented a specific capacity of 210 mA h g$^{-1}$ at 20 mA g$^{-1}$, and 136 mA h g$^{-1}$ was retained after 2000 cycles at 20 C [54]. $Ti_{0.94}Nb_{0.06}O_2$ thick-film anodes exhibited a reversible capac-

ity of 160 mA h g$^{-1}$ at 50 mA g$^{-1}$ upon the 50th cycle as an anode for SIBs [55]. He et al. fabricated Fe-doped cauliflower-like rutile TiO$_2$ for Na storage [56]. The Fe doping not only enhanced the conductivities, but also created more oxygen vacancies and lowered the sodiation energy barrier. This TiO$_2$-based electrode exhibited outstanding cycling stability, maintaining a specific capacity of 127.3 mA h g$^{-1}$ over 3000 cycles at 1680 mA g$^{-1}$.

The sodium storage mechanism occurring in anatase could be explored using high-energy X-ray diffraction [57]. In Figure 2a, the pristine anatase TiO$_2$ was in good order, with Na accommodating in 3a Wyckoff sites and Ti occupying in 3b. After full sodiation, the 3a sites were filled with 43% Na and 57% Ti, and 39% Na and Ti took up part of the 3b sites. The chemical formula could be expressed as (Na$_{0.43}$Ti$_{0.57}$)$_{3a}$($V_{0.22}$Na$_{0.39}$Ti$_{0.39}$)$_{3b}$O$_2$, ($V$ is vacancy), highlighting the strong structural disorder derived from cationic intermixing. Compared with NaTiO$_2$, a contraction of the c parameters occurred in this situation. Upon desodiation, the structure was no longer a layered compound but a local structure similar to the anatase TiO$_2$.

In the consideration of anodes for batteries, a low-operation voltage is preferable in order to achieve a high energy density for practical utilization. Na-Ti-O composites with different Na/Ti ratios, such as NaTiO$_2$, Na$_2$Ti$_6$O$_{13}$, and Na$_4$Ti$_5$O$_{12}$, especially Na$_2$Ti$_3$O$_7$, have been widely studied [58–61]. The first proposal for the application of Na$_2$Ti$_3$O$_7$ in SIB was put forward by Senguttuvan et al. in 2011 [62]. It exhibits a low average voltage potential of about 0.3 V, which is only a little higher than that for carbon and NaPb alloys (as shown in Figure 2b). Na$_2$Ti$_3$O$_7$ could accommodate 2 Na$^+$ reversibly (2 Na + Na$_2$Ti$_3$O$_7$ $\rightleftarrows$ Na$_4$Ti$_3$O$_7$) with full sodiation and achieve a specific capacity of 200 mA h g$^{-1}$ at C/25. Detailed studies with the aid of analysis of electrochemical behavior and DFT calculations found that the inserted Na$_4$Ti$_3$O$_7$ phase is mechanically and dynamically stable, indicating that the insertion reaction is fully reversible between Na$_2$Ti$_3$O$_7$ and Na$_4$Ti$_3$O$_7$ [63]. Surface modification as well as novel morphologies could provide assistance to enhance the performance of Na$_2$Ti$_3$O$_7$ [64–66]. For instance, Na$_2$Ti$_3$O$_7$ with 3D hierarchical flower-like stable microstructures was synthesized by the hydrothermal process [67]. Interlaced 2D Na$_2$Ti$_3$O$_7$ nanosheets closely interlinked with each other to form carnation-shaped 3D Na$_2$Ti$_3$O$_7$ (Figure 2c,d). The layered 3D microstructures with a high surface area and effective pore size demonstrated an impressive long-term cycling life (Figure 2e), retaining a specific capacity of 85 mA h g$^{-1}$ at 400 mA g$^{-1}$ even after 1100 cycles (80% of initial specific capacity). At 800 mA g$^{-1}$, a discharge-specific capacity of 73.8 mA h g$^{-1}$ was still obtained, indicating the enhanced rate capability.

Different from the generally accepted Na$_2$Ti$_3$O$_7$ $\rightleftarrows$ Na$_4$Ti$_3$O$_7$ mechanism, a fresh Na$_2$Ti$_3$O$_7$ $\rightleftarrows$ Na$_{3-x}$Ti$_3$O$_7$ pathway delivering a redox voltage of 0.2 V vs. Na/Na$^+$ was unveiled by Rudola et al. [68]. The distinct voltage step implied that an intermediate phase formed between Na$_2$Ti$_3$O$_7$ and deep discharged Na$_4$Ti$_3$O$_7$ between 2.5 and 0.01 V. In the discharge process, the ex situ XRD results clearly illustrated that the intermediate Na$_{3-x}$Ti$_3$O$_7$ formed at first (from A to C) and then gradually decreased (from D to E), and only Na$_4$Ti$_3$O$_7$ existed at the fully discharged state. With this route, Na$^+$ storage along the lower discharge plateau underwent a two-phase reaction in nature. Thus, simply limiting the cut-off voltage around 0.155 V (which was called shadow discharge) made the cell exhibit different electrochemical properties. The discharge/charge voltages are 0.24/0.22 vs. Na/Na$^+$ other than 0.47/0.44 V in deeper discharge.

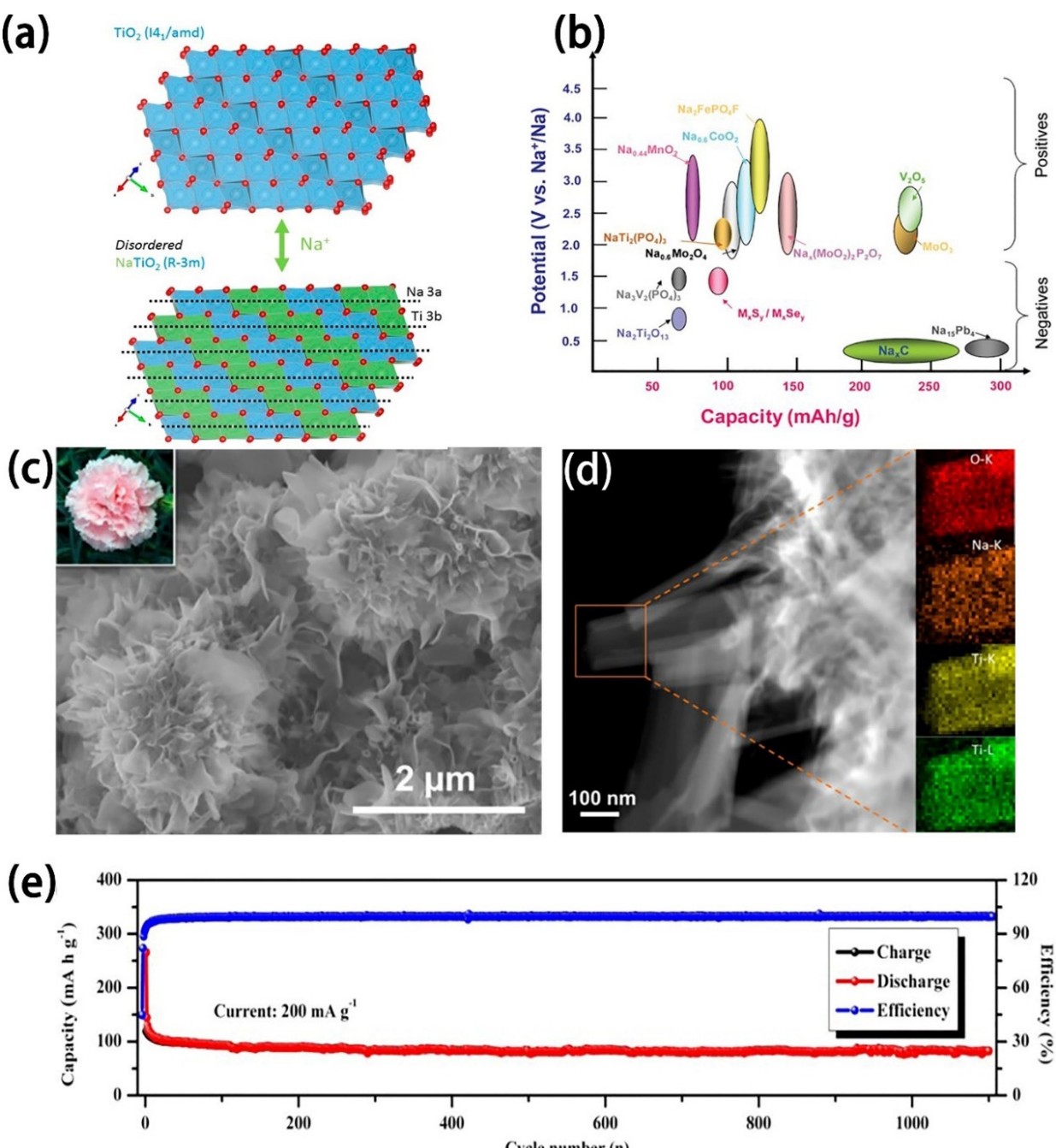

**Figure 2.** (**a**) Structural relationship between the pristine tetragonal and sodiated rhombohedral phases along the (100) direction of anatase $TiO_2$. Reproduced with permission of [57]. Copyright 2017, American Chemical Society. (**b**) Voltage capacity plot of various materials for potential electrode materials for sodium-ion batteries. Reproduced with permission of [62], Copyright 2011, American Chemical Society. (**c**) Low-magnification SEM images; (**d**) scanning transmission electron microscopy (STEM) high-angle annular dark field (HAADF) image as well as (**e**) the long cycle performance measured at a current density of 400 mA g$^{-1}$ of $Na_2Ti_3O_7$ nanostructures. Reproduced with permission of [67]. Copyright 2017, American Chemical Society.

### 2.2.2. Other Conversion-Type Oxides

Compared with Ti-based anodes, the sodium storage process of other transition metal oxides (TMOs) is usually based on a conversion mechanism, which delivers much higher capacities. Nevertheless, the drawbacks of a large volume variation during charg-

ing/discharging and inherent sluggish electron transport would lead to an unsatisfactory reversible capacity. Analogously, a lot of well-known anodes for LIBs already utilized in SIBs still demonstrate attractive performances in SIBs. Hariharan et al. first reported $\alpha$-$MoO_3$ anode materials for sodium storage in 2013 [69]. The rocking chair $MoO_3$/$Na_3V_2(PO_4)_3$ full cell revealed a plateau at about 1.4 V, and the obtained initial discharge-specific capacity was 164 mA h $g^{-1}$, based on anode weight. Moreover, coupled with a carbon-coated $Na_3V_2(PO_4)_3$ cathode, the average voltage of a $V_2O_5$/$Na_3V_2(PO_4)_3$ full cell occurs at 2.5 V, and the specific capacity of 113 mA h $g^{-1}_{anode}$ could be maintained after cycling 200 times at 100 mA $g^{-1}$ [70]. With the adoption of a carbon-$Fe_3O_4$/$Na_2FeP_2O_7$ configuration, the full cell also delivered a satisfactory specific capacity of nearly 100 mA h $g^{-1}$ and a distinct voltage platform of about 2.28 V, which equals an energy density of 203 W h $kg^{-1}$ [71]. In addition, other components in batteries also influence the practical capacity of full batteries, including the electrode's formula, balance, binders, and electrolytes [72,73].

Similar to the situation of LIBs, although Ti-based materials have the merit of good structural stability, their relatively low specific capacity of about 100–200 mA h $g^{-1}$ is generally far less than their theoretical capacities. In contrast, the remarkable advantage of TMOs is their high theoretical capacity. But, further exploration of the approach to alleviating the large volume variations should be highlighted due to the poor cycling performance.

## 2.3. Intermetallic Compounds

Electrodes based on Si, Ge, and Pb, which deliver high reactivity in LIBs, have a smaller or no specific capacity for SIBs, while Sn reacts with Na to reversibly form Sn-Na intermetallic phases and display electrochemical redox reactions [74]. It could alloy with up to 3.75 Na to acquire a theoretical specific capacity of 847 mA h $g^{-1}$ and an average voltage around 0.3 V [75]. But the key problem is that the fast capacity fading resulted from the inherent severe volume change of 420% (as shown in Figure 3a [76]). One approach to relieve the volume variation is to insert Sn nanoparticles in a highly conductive matrix that mostly consists of carbonaceous material [77]. A novel Sn/N-doped carbon microcage was prepared by a facile spray-drying strategy [78]. The authors adopted ethylenediaminetetraacetic acid (EDTA) as the carbon and nitrogen source, and NaCl acted as the pore generator. With its delicate design, the Sn/C composite exhibited an open-framework structure, and the buffer matrix could alleviate stress while accommodating for volume changes during cycling. Oh et al. assembled an Sn-C/$Na(Ni_{0.5}Mn_{0.5})O_2$ full cell, which operated reversibly around 2.8 V within 1.5–4.0 V at 24 mA $g^{-1}$ [79]. Another porous carbon nanocage (PCNCs-Sn) encapsulating with Sn nanoparticles was developed by template-assisted CVD and an in situ reduction route [80]. This anode exhibited a good high-rate capacity of 188 mA h $g^{-1}$ at 2560 mA $g^{-1}$ (about 3 C) and superior cycling performance even after 1000 cycles at 2560 mA $g^{-1}$ in a half-cell. The novel carbon nanocages and well-preserved porous structure effectively inhibited the volume variation, and the weak bonds formed between graphitic carbon and the discharge product ($Na_{15}Sn_4$) also maintained the superior conductivity, reducing the pulverization. After coupling PCNCs-Sn with the P2-$Na_{0.80}Li_{0.12}Ni_{0.22}Mn_{0.66}O_2$ cathode, the advanced full SIB could sustain a red-light-emitting diode (LED) working for two days.

Reducing Sn particle size is also effective for alleviating the strain. For example, the porous Sn anode prepared via a phase-inversion technique enabled the free expansion of Sn particles [81]. Electrode degradation was inhibited to some extent, improving long-term cycling stability. Correspondingly, the Sn electrode delivered a reversible capacity of more than 519 mA h $g^{-1}$ after 500 cycles. However, along with the adequate free space for volume change offered by the nanostructure, a low density and high irreversible capacity also come out.

In addition, significant differences would occur when using different electrolytes, which are decided by the different properties of the formed SEI film. Zhang has attempted to apply different electrolytes for SIBs with micro-sized Sn particles as the anode [82]. They found that the surface is smoother in glyme-based electrolytes than carbonate-based ones.

And the micro-sized Sn//1 M NaPF$_6$ in DGME//Na$_3$V$_2$(PO$_4$)$_3$/C full battery displayed a gravimetric energy density of 200 W h kg$^{-1}$ and a volumetric energy density of 703 W h L$^{-1}$, which is much higher than that with 1 M NaPF$_6$ in the EC/DMC electrolyte. The result highlights the benefit of using Sn as an anode.

Sb with a high theoretical specific capacity of 660 mA h g$^{-1}$ is another attractive anode candidate when fully sodiated to Na$_3$Sb [83]. By directly using commercially available Sb powders or beads as the raw materials, Sb/graphite was obtained by a simple ball-milling process [84]. The full cells constructed by the Sb/graphite anode and Na$_{1.72}$Mn$_{0.72}$Fe(CN)$_6$ cathode presented a markedly reduced polarization. Normalized by the active mass of Na$_{1.72}$Mn$_{0.72}$Fe(CN)$_6$, the full cell delivered initial discharge/charge capacities of 115/155 mA h g$^{-1}$ with the corresponding Coulombic efficiency of 74% (Figure 3b,c). Zhang reported a kind of Sb/rGO paper electrode with no utilization of the binder, conducting additives, or metal current collectors [85]. Coupled with the Na$_3$V$_2$(PO$_4$)$_3$/rGO paper cathode, the reversible specific capacity of the full cell could achieve 400 mA h g$^{-1}$ at 100 mA g$^{-1}$ after 100 cycles. The idea of all free-standing electrodes offers giant potential for bendable electrodes in the field of smart electronics, especially wearable devices.

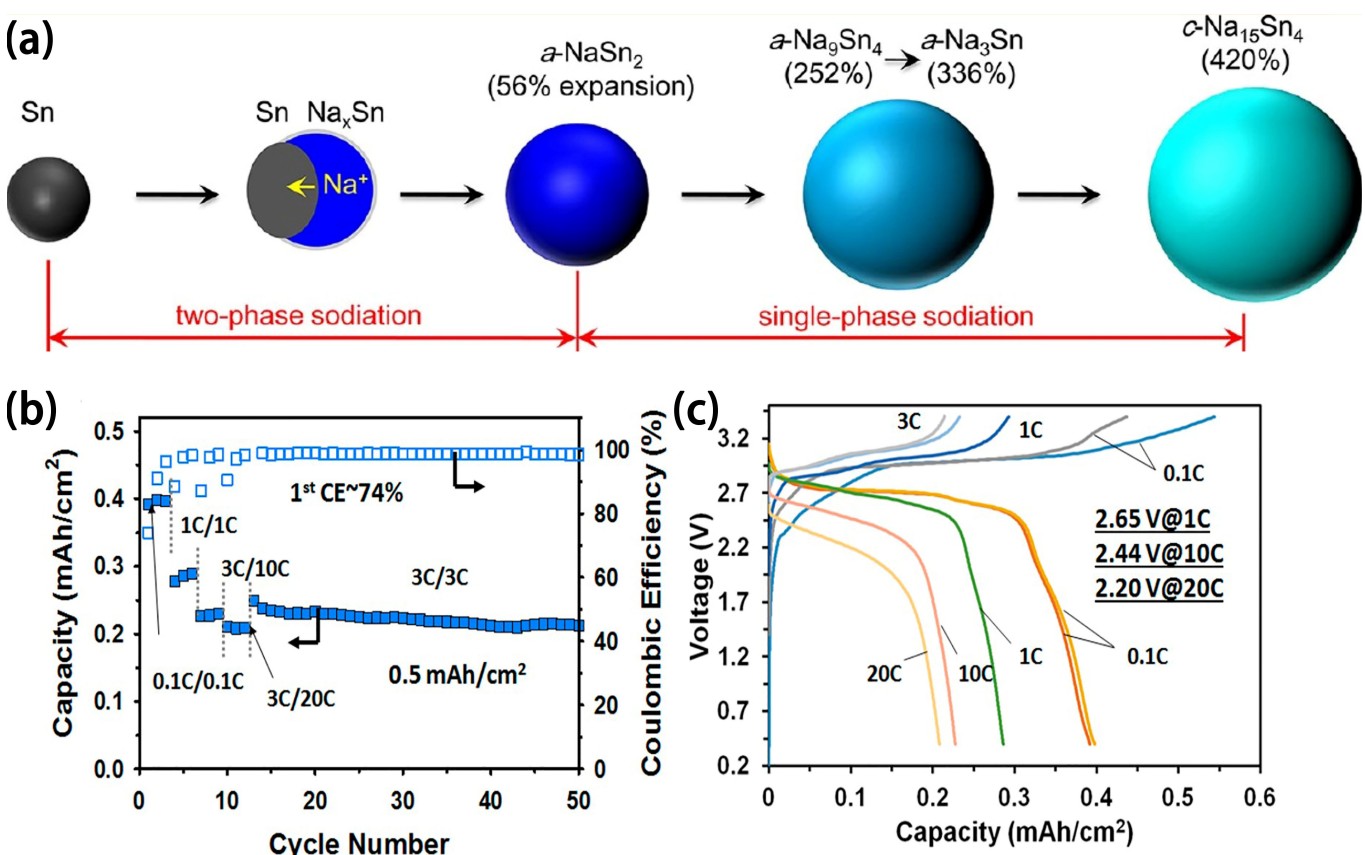

**Figure 3.** (**a**) Structural evolution process of Sn nanoparticles during sodiation. Reproduced with permission of [76]. Copyright 2012, American Chemical Society. (**b**) Specific desodiation capacity and Coulombic efficiency at 0.1–20 C and (**c**) galvanostatic charge/discharge profiles of Sb/graphite/Na$_{1.72}$Mn$_{0.72}$Fe(CN)$_6$ full cells. Reproduced with permission of [84]. Copyright 2016, American Chemical Society.

Apart from hybridizing with carbon materials, oxygen-deficient TiO$_2$ (TiO$_{2-x}$) offers another opportunity to exert a positive influence on the performance of Sb anodes [86]. The double-walled Sb@TiO$_{2-x}$ nanotubes combined the merits of outstanding structural stability of TiO$_{2-x}$ and satisfactory capacity of Sb, leading to outstanding electrochemical

properties, both in LIBs and SIBs. In a Sb@TiO$_{2-x}$//Na$_3$V$_2$(PO$_4$)$_3$-C full cell, the potential was about 2.5 V; when cycled at 660 mA g$^{-1}$$_{anode}$ after 100 cycles, a specific capacity of 350 mA h g$^{-1}$$_{anode}$ was maintained. The energy density was 151 W h kg$^{-1}$ at 21 W kg$^{-1}$, or 61 W h kg$^{-1}$ at 1.83 kW kg$^{-1}$. Other intermetallic compounds, such as transition metal selenides could also be explored as viable SIB anodes [87,88]. The intermetallic compounds are suitable choices for the commercial application of SIBs due to the combination of their respective merits in the whole material, including their high capacity and low working potential.

### 2.4. Sulfides and Phosphates

Transition metal chalcogenides have raised some concerns because of their two-dimensional layered structure analogous to graphite and the high capacities induced by multiple electrons involved in the conversion reaction. However, several general issues still hinder their application. First, the majority of transition metal chalcogenides suffer from poor electrical conductivity, requiring composites with other high-conductive materials such as carbon. Second, the typical conversion mechanism triggers large volume variation during repeated sodiation/desodiation, deteriorating the cycling performance [89]. Third, the discharge products of sodium sulfide tend to dissolve in most electrolytes, leading to a significant mass loss of active materials. Furthermore, the limited interlayer distance between the adjacent layers also blocks the Na$^+$ insertion without further modification.

MoS$_2$ with a high theoretical specific capacity of 670 mAh g$^{-1}$ is of particular interest [90,91]. Despite the fast Na$^+$ diffusion in 2D permeable channels, the poor conductivity, aggregation, and the inadequate interlayer distance of MoS$_2$ monolayers are still to be solved [92]. Atomic interface engineering was carried out to modify the layered MoS$_2$ with smart morphology [93]. A hierarchical MoS$_2$/carbon nanotube superstructure consisting of MoS$_2$ and inserted carbon layers was prepared by a one-pot solvothermal synthesis; the interlayer distance between adjacent MoS$_2$ layers was enlarged to 0.986 nm, which is larger than 0.615 nm of the bulk pure MoS$_2$. Although the interlayer distance between MoS$_2$ and the inserted carbon layer was only 0.495 nm, the carbon layers were not continuous according to the calculation. By virtue of conductive carbon and enlarged interface spacing for easier Na$^+$ diffusion, the half-cell of the MoS$_2$/carbon sandwiched structure achieved an initial discharge-specific capacity of 620 mAh g$^{-1}$ with a first cycle Coulombic efficiency of 84%. As early as 1979, Whittingham's group proposed that MoS$_3$ could react with sodium to form Na$_4$MoS$_3$ [94]. Ye et al. verified that amorphous MoS$_3$ seems to be better than crystalline MoS$_2$ in electrochemical performance [95]. Compared with the well-studied 2D layered MoS$_2$, the amorphous MoS$_3$ is composed of 1D Mo chains bridged by sulfide and disulfide ligands [96], which reaches a 25% higher theoretical capacity. This structure is conductive to the rapid Na$^+$ diffusion, and the amorphous nature requires a much lower activation energy barrier during the structural rearrangement; thus, an improved electrochemical performance of about 615 mAh g$^{-1}$ at 50 mA g$^{-1}$ and 235 mAh g$^{-1}$ at 20 A g$^{-1}$ was achieved. Furthermore, after pairing the amorphous MoS$_3$ anode with the Na$_3$V$_2$(PO$_4$)$_3$ cathode, the full cells tested between 0.1–3.0 V exhibited an average voltage of ≈1.8 V, and a reversible specific capacity of 520 mAh g$^{-1}$ was retained at 50 mA g$^{-1}$. Even after 100 cycles, the delivered specific capacities were 415 and 376 mAh g$^{-1}$ at 500 and 1000 mA g$^{-1}$, respectively.

Ni-based sulfides with disparate Ni/S ratios such as NiS, NiS$_2$, Ni$_3$S$_2$, Ni$_3$S$_4$, Ni$_7$S$_6$, and Ni$_9$S$_8$ are also attractive due to the high electrochemical activity of nickel ion and the strong reducibility of sulfur ion [97–99]. The theoretical specific capacity of NiS$_2$ is as high as 879 mAh g$^{-1}$. Like other transition metal chalcogenide anodes, NiS$_2$ also undergoes severe volume expansion and structure collapse during cycling. The introduction of graphene nanosheets played an important role in alleviating volume change, serving as a conductive matrix for NiS$_2$ and relieving the aggregation of NiS$_2$ nanoparticles [100]. Qin et al. also introduced rGO into Ni$_3$S$_2$ [101], and the Ni$_3$S$_2$ content tended to decrease while tiny Ni$_7$S$_6$ appeared along with the increased content of rGO in the Ni$_3$S$_2$/rGO composite, which was likely due to the different nucleation processes with and without rGO. The cell exhibited a

reversible specific capacity of 391.6 mAh g$^{-1}$ at 100 mA g$^{-1}$ after 50 cycles. Apart from the Mo- and Ni-based sulfides mentioned above, there are also diverse transition metal chalcogenides including $Sb_2S_3$, $ZnS$, $SiS$, $V_5S_8$, $SnS_2$, $WS_x$, and so on [102–104].

Titanium phosphates $NaTi_2(PO_4)_3$ (NTP) with a theoretical specific capacity of 133 mAh g$^{-1}$ and voltage potential of 2.1 V vs. $Na^+/Na$ are excellent anode candidates in large-scale applications [105]. Similar to other NASICON-type materials, it has the distinct merit of high $Na^+$ conductivity. NTP@C nanocomposites prepared by the facile sol–gel method deliver outstanding $Na^+$ storage and cycling stability [106]. The highly crystalline NTP particles (15–50 nm in size) were covered by a dense carbon layer of about 7 nm in thickness followed by embedding inside the carbon matrix. By widening the cutoff voltage to 0.01–3V, an extra platform appeared at around 0.4 V, which originated from a further reduction from $Ti^{3+}$ to $Ti^{2+}$. Remarkably, a capacity of about 68% still retained even cycling over 10,000 times at 4 A g$^{-1}$, denoting an ultralong cycle life. In addition, the NTP/C anode synthesized by Jiang et al. delivered an outstanding reversible specific capacity of 108 mAh g$^{-1}$ even at 100 C, and a specific capacity of 83 mAh g$^{-1}$ was retained, even lasting for over 6000 cycles at 50 C. [107]. These results clearly prove that NTP@C nanocomposites possess a stable structure that could bear repeated sodiation/desodiation, suitable for devices, especially those requiring a long lifespan. Therefore, coupling with $Na_3V_2(PO_4)_3/C$ cathode [108], the full cell released a discharge-specific capacity of 128 mAh g$^{-1}_{anode}$ at 0.1 C, and excellent rate performances, retaining a specific capacity of 88 mAh g$^{-1}$ at 50 C. After 1000 cycles at 10 C, 80% of the initial capacity is retained, indicating the high cycling stability. However, the relatively higher working potential of 2.1 V would reduce the whole energy density of full cells. In addition, $Na_3V_2(PO_4)_3$ (NVP) is able to act as a bipolar electrode material in symmetric full batteries.

### 2.5. Organic Compounds

Compared with inorganic electrodes, organic compounds are appealing due to their low cost, flexible designability, and sustainability [109]. Most of the present organic electrodes are carbonyl derivatives, typically $Na_2C_8H_4O_4$, and the others reported in recent years include biomolecular-based and Schiff-based compounds [110,111].

The first application of organic compounds in batteries was proposed by Armand, using di-lithium terephthalate $Li_2C_8H_4O_4$ with two carboxyl groups as the anode for LIBs [112], which exhibited a low potential of 0.8 V with 2.3 Li insertion per unit. Then, $Na_2C_8H_4O_4$, as the analogue of $Li_2C_8H_4O_4$, conjugated carboxylate organic molecules containing carbon–oxygen double bonds (C=O) undergo a reversible conversion between C=O and C-O in the Na ion embedding/escaping process, was introduced into a $Na^+$ ion system [113]. The discharge/charge profiles of $Na_2C_8H_4O_4$ between 0.1 and 2 V reveal that Na insertion voltage occurs at 0.29 V vs. $Na^+/Na$ and a reversible specific capacity of 250 mAh g$^{-1}$ accompanying a two electron transfer. Moreover, the pioneering work by Zhu et al. employed 2, 5-dihydroxy-1 and 4-benzoquinone ($Na_2C_6H_2O_4$) ($Na_2DBQ$) as negative electrodes for SIBs [113]. $Na_2DBQ$ was synthesized by a substitution reaction between $H_2DBQ$ and sodium methylate in DMSO. The two carbonyl groups in $Na_2DBQ$ are redox centers that are able to absorb two sodium ions, delivering a high theoretical specific capacity of 291 mA h g$^{-1}$. The as-prepared $Na_2DBQ$ could reach a practical specific capacity of 265 mAh g$^{-1}$ at 0.1 C. By increasing the current density to 5 C, the specific capacity of 160 mA h g$^{-1}$ is retained. Remarkably, the specific capacity stabilized around 181 mAh g$^{-1}$, even after 300 cycles at 1 C. To further understand the mechanism of organic carbonyl compounds, $Na_2C_6H_2O_4$ was selected to test the sodium storage behavior [114,115]. It was found that the inorganic Na-O polyhedral layer offers channels for $Na^+$ diffusion, and the organic p-stacked benzene layer facilitates rapid electron transfer.

Natural-derived redox biomolecules are deemed good candidates in view of their advantages of biodegradablity, abundance, being lightweight, and being easily processed. For example, since juglone could stem from waste walnut epicarp, the aluminum pouch full SIB with juglone/rGO film anode and $Na_3V_2(PO_4)_3/C$ electrode delivered a reversible

discharge-specific capacity of 80 mA h g$^{-1}$ with well-defined redox working potentials of 1.3/1.85 V [116].

### 2.6. 2D Materials

Two-dimensional (2D) materials demonstrating novel structural architectures could exert an important influence on sodium-ion diffusion by providing abundant ion transport channels and a low activation barrier. Moreover, the outstanding mechanical properties together with the unique flexibility endow it with high endurance to both stress and strain, which makes it very suitable for the application as electrode materials with high capacities (Figure 4a) [117]. Currently, various materials have been explored, including transition metal carbides and nitrides (MXenes), black phosphorus, as well as transition metal oxides (TMOs) [118,119]. Typically, transition metal carbides and nitrides (MXenes) as one of the burgeoning anode materials exhibit excellent electrical conductivity and surface hydrophilicity [120]. For example, an optimized hybrid electrode composed of antimony/MXene has been obtained by delicate designs on synthesis parameters [121]. The homogeneous distribution of antimony and MXene jointly contributes to a high reversible specific capacity of 450 mA h g$^{-1}$ at 0.1 A g$^{-1}$ with a capacity retention of around 96% after 100 cycles, as well as 365 mA h g$^{-1}$ at 4 A g$^{-1}$. Moreover, thermodynamically stable black phosphorus (BP) with good electric conductivity (about 300 S m$^{-1}$) exhibits a double-layer structure with a channel size of 0.308 nm, which is conductive to the access for lithium ions transport. And its theoretical specific capacity could achieve as high as 2596 mA h g$^{-1}$ [122]. However, volume variation during cycling would cause the structural pulverization of BP, and the easy dispersion in the electrolyte leads to the loss of electronic contact with the current collector, resulting in severe capacity fading. To solve this critical issue, a core–shell heterostructure composed of red and black phosphorus on a 3D N-doped graphene framework could be synthesized by the one-pot solvothermal method [123]. The extremely high electronic conductivity together with the low sodium-ion diffusion barrier leads to an ultra-high areal capacity of 3.46 mA h cm$^{-2}$, corresponding to a reversible specific capacity as high as 1440.2 mA h g$^{-1}$ at 0.05 A g$^{-1}$. And the capacity retention could be retained at 89.3% even after 1200 cycles at a high current density of 10 A g$^{-1}$.

In addition, preparing hybrid 2D materials is another effective way to improve the electrochemical performance of BP. For instance, H. Liu et al. bridged covalently functionalized BP on graphene to achieve significantly enhanced reversible performance [124]. Benefiting from the excellent electrical connection provided by chemical bonds between the rGO slice and modified BP nanosheet, the BP-based composite anode is able to release a specific capacity of 1472 mA h g$^{-1}$ at 0.1 A g$^{-1}$ after 50 cycles as well as 650 mA h g$^{-1}$ at 1 A g$^{-1}$ after 200 cycles (Figure 4b,c). Moreover, the combination of black phosphorus quantum dots (BPQDs) and $Ti_3C_2$ nanosheets (TNSs) exhibits an enhanced electronic conductivity, and the stress during cycling could be relived significantly, resulting in stable and satisfactory sodium storage performance [125]. Meanwhile, a high pseudocapacitive value induced by P-O-Ti interfacial bonds could also be obtained. Very recently, a MXene-$MoS_2$ heterostructure prepared by T. Wang and co-workers combined the merits of high electronic conductivity, excellent structural stability, as well as reduced ion diffusion length [126]. And the final composite, as an anode for SIBs, delivered 315 mA h g$^{-1}$ at 0.2 A g$^{-1}$ and 220 mA h g$^{-1}$ after 1000 cycles at 2.0 A g$^{-1}$.

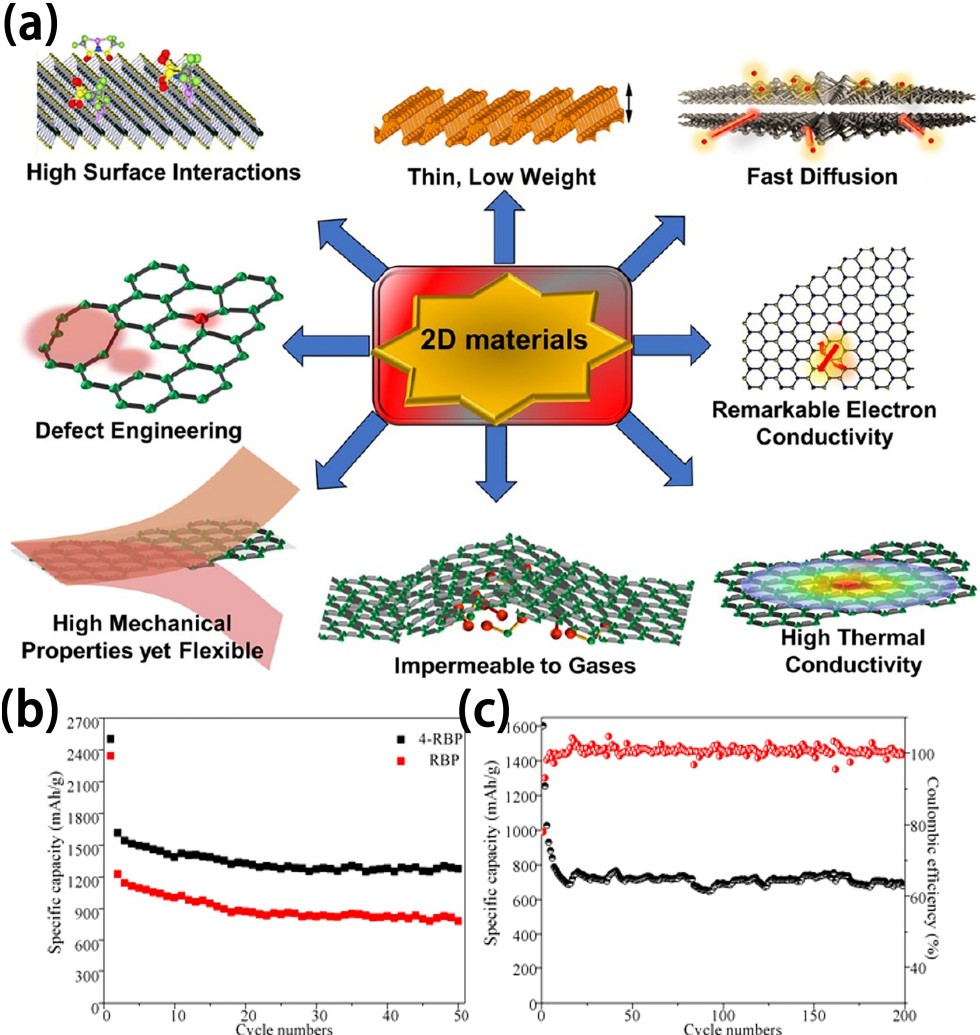

**Figure 4.** (**a**) The advantages of 2D materials applied in energy storage devices. Reproduced with permission of [117]. Copyright 2020, American Chemical Society. (**b**) Specific capability of 4-nitrobenzenediazonium modified black phosphorus chemically bonded with reduced graphene oxide hybrid (4-RBP) and black phosphorus bonding with RGO (RBP) at current density of 0.1 A g$^{-1}$. (**c**) Specific capability and Coulombic efficiency of 4-nitrobenzenediazonium modified black phosphorus chemically bonded with reduced graphene oxide hybrid (4-RBP). Reproduced with permission of [124]. Copyright 2017, American Chemical Society.

## 3. Prospect

Due to their abundance on the earth, their low cost, and similar intercalation mechanisms with LIBs, SIBs appear to be a very competent substitution for LIBs. However, as evidenced from the reported literature, the electrochemical performance, especially the reversible specific capacity, cannot keep up with that of LIBs. Searching for better electrode materials and carefully designing balanced electrodes indicate their great importance. In this review, the current categories of anode materials and the electrochemical performance for both half and full cells have been summarized. Although the specific capacity and energy density have been greatly enhanced, some issues should also be paid attention to, as follows: (1) The transport of both ions and electrons needs to be further improved for all the anode materials to improve the reaction kinetics. (2) Adopting cost-effective methods to obtain appropriate anode materials is more favorable to the commercialization of SIBs, but this is still a challenge since the novel morphology and multi-step modifications progress usually requires a long period and expensive devices. (3) Safety still acts as the key focus of attention. Correspondingly, the application of SIBs at extreme working conditions,

especially at high and low temperatures, should pay close attention to the modification of electrolytes with non-inflammability and wide voltage windows, except for the design of unique structures of anode materials. Therefore, it is believed that there is still a long way to go to push SIBs into mature markets before solving the above-mentioned issues.

**Author Contributions:** Conceptualization, X.B. and T.L.; methodology, X.B.; validation, T.L. and N.W.; visualization, N.W. and G.Y.; writing—original draft preparation, X.B.; funding acquisition, T.L. All authors have read and agreed to the published version of the manuscript.

**Funding:** This research was funded by Shandong Provincial Natural Science Foundation, China (Grant No. ZR2022QE181).

**Data Availability Statement:** Not applicable.

**Acknowledgments:** The authors would like to thank the Shandong Provincial Natural Science Foundation, China (Grant No. ZR2022QE181).

**Conflicts of Interest:** The authors declare no conflict of interest.

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
