# Peer review of "Recent Advances in Anode Materials for Sodium-Ion Batteries"

_inorganics, doi:10.3390/inorganics11070289_

Round 1

Reviewer 1 Report

This paper reviews recent studies on anode materials of sodium-ion batteries (SIBs). Overall the review is well prepared with authors’ view on the prospect of SIB anode. The current manuscript is acceptable for publication. My recommendation is to renew the references. There has been extensive research on SIBs in the past few years. This review, however, doesn’t include lots of the latest studies. Only 4~5 references published in 2021 and 2022 have been included.

Acceptable and easy to understand.

Author Response

Thanks for your suggestions. The references have been renewed based on your advice and the corresponding content has been revised in the revised manuscript.

Reviewer 2 Report

No major comments, the article can accept in present form Figure 3 b need to improve the quality. I adivce to add more recent literature between 2020 to 2022.

No major comments, the article can accept in present form Figure 3 b need to improve the quality. I adivce to add more recent literature between 2020 to 2022.

Author Response

Thanks for your suggestions. The quality of Figure 3 has been improved, and the references have been renewed based on your advice as well. The corresponding content has been revised in the revised manuscript.

Reviewer 3 Report

The review article nicely summarizes the “Recent advances on anode materials for sodium ion batteries”. The larger ionic radius of Na+ (as compared with Li+) could lead to slower reaction kinetics, resulting in poor reversible capacity and rate capability. Therefore, the main concern is to search, design and implement new anode materials that can accommodate Na+ reversible without affecting the overall performance of the electrode.

The review article has touched on various anode materials and associated issues and proposed solutions that have been reported in the literature. There has been a new development in exploring more flexible batteries by utilizing two-dimensional materials. The authors well documented and explained the role of metal oxides, transition metal oxides and sulfides as anode materials. But considering a growing demand for more flexible batteries: MXene and black phosphors can also be applied as an anode system to boost the electrochemical performance.

I would advise the authors to add more applications on MXenes, and black phosphorus. Even the trend is to have more hybrid anode materials toward target functionality. Individual 2d materials also have several drawbacks such poor electrical conductivity, scalability and layer stacking that can affect the performance of sodium ion batteries.

A detailed discussion on the role of 2d materials should be added in the introduction as well as separate sections for MXene, BP and the hybrid anodes with application should be included in this review article. I would also suggest authors to include latest review articles in the references:

1 1.  Rojaee, R., and Shahbazian-Yassar, R. 2020. Two-Dimensional Materials to Address the Lithium Battery Challenges. ACS Nano. 14: 2628-2658.

   2. Xiao, Z., et al. 2021. Recent Developments of Two-Dimensional Anode Materials and Their Composites in Lithium-Ion Batteries. ACS Appl. Energy Mater. 4: 7440-7461.

3 3.  Li, Y. et al. 2019. Emerging two-dimensional noncarbon nanomaterials for flexible lithium-ion batteries: opportunities and challenges, J. Mater. Chem. A. 7: 25227.

Author Response

Thanks for your suggestions. The discussion about MXene, BP and the hybrid anodes have been added in the revised manuscript, and the latest articles has also been added.

Round 2

Reviewer 3 Report

The review article looks more compact by adding the discussion about 2D materials such as MXene, BP and the hybrid anodes in the revised manuscript. This manuscript is now ready for publication in Inorganics.

Author Response

Thanks for your comments.